

# Reconstruction of mass balance and firn stratigraphy during the 1996-2011 warm period at high-altitude on Mt. Ortles, Eastern Alps: a comparison of modelled and ice core results

Luca Carturan[1], Alexander C. Ihle[2,3], Federico Cazorzi[4], Tiziana Lazzarina Zendrini[1], Fabrizio De Blasi[1,5], Giancarlo Dalla Fontana[1], Giuliano Dreossi[6], Daniela Festi[7], Bryan Mark[3], Klaus Dieter Oeggl[8], Roberto Seppi[9], Barbara Stenni[6], Paolo Gabrielli[10]

[1]Department of Land, Environment, Agriculture and Forestry, University of Padua, Legnaro, Italy

[2]Department of Earth and Environmental Sciences, University of Rochester, Rochester, USA

[3]Department of Geography and Byrd Polar and Climate Research Institute, Ohio State University, Columbus, USA

[4]Department of Agricultural, Food, Environmental and Animal Sciences, University of Udine, Udine, Italy

[5]Consiglio Nazionale delle Ricerche - Istituto di Scienze Polari, c/o Ca' Foscari University of Venice, Venice, Italy

[6]Department of Environmental Sciences, Informatics and Statistics, Ca' Foscari University of Venice, Venice, Italy

[7]GeoSphere Austria, Department of Geoanalytics and Reference Collections, Vienna, Austria

[8]Department of Botany, University of Innsbruck, Innsbruck, Austria

[9]Department of Earth and Environmental Sciences, Pavia, Italy

[10]Italian Glaciological Committee, Torino, Italy

*Correspondence to:* Luca Carturan (luca.carturan@unipd.it)

**Abstract**

Paleoclimatic glacial archives in low-latitude mountain regions are increasingly affected by melt, which leads to heavy percolation and can remove snow and firn accumulated across months, seasons or even years. Proxy system models, used for improved interpretation of glacial proxies and paleoclimatic reconstructions, generally do not account for melt because they are optimized for sites where snow layer removal by melting is negligible. In this paper, we present a mass balance model applied to the Mt. Ortles drilling site, at 3859 m a.s.l. in the Eastern Italian Alps, with the aim of building a pseudo proxy of atmospheric conditions during the formation of snow layers survived to ablation. This pseudo proxy is useful for improved dating and environmental interpretation of firn layers (<15 m depth), affected by significant melt in the period 1996-2011, which includes the extremely warm summer 2003. Here we show that the model significantly improves



the interpretation of the firn stratigraphy. This is fundamental for detecting melted layers and for refining the dating of
the core based on traditional annual layer counting of stable isotope and pollen seasonal oscillations.

**1 Introduction**
Atmospheric warming is threatening paleoclimatic glacial archives, particularly those located in low-latitude mountain
areas (e.g. Gabrielli et al., 2010; Huber et al., 2024). When compared to polar ice sheets in Greenland and Antarctica,
these glaciers are now altitudinally closer to their lower limits of formation and preservation.
Long before their complete disappearance, glacial archives are affected by post-depositional processes caused by
increasing temperature, which modifies and ultimately overprints their paleoclimatic signals. Increasing frequency and
intensity of surface melt events lead to snow/firn layer mass loss and heavy percolation of meltwater through the firn.
This obliterates part of the glacial archive and smooths or dislocates intra/inter-annual variations of chemical impurities
of interest for paleoclimatic reconstructions (Dietermann & Weiller, 2013; Gabrielli et al., 2010; Hashimoto et al., 2005;
Lee, 2014; Moran et al., 2011; Moser et al., 2024, Thompson et a., 2011, 2021; Unnikrishna et al., 2002).
It is unclear how extreme melt events, such as the summer 2003 heat wave in the European Alps (Zappa and Kan, 2007;
García-Herrera et al., 2010), affect the preservation of ice core archives. This kind of events may significantly change the
original isotopic record, melting the snow accumulated over several months or years (Gabrielli et al. 2010). In addition it
is unknown whether such extreme events may relocate less mobile impurities such as pollens or black carbon, as their
annual cycle is generally preserved under melting conditions, and are therefore used for ice core dating (Pavlova et al.,
2015; Festi et al., 2021; Takeuchi et al., 2019; Moser et al., 2024).
Atmospheric warming also affects snow water content and its metamorphism, which control snow redistribution by wind.
For this reason, snow drifting is expected to be most effective for cold and dry snow and less so for wet snow and melt-
freeze crusts (Haeberli and Alean, 1985, Li and Pomeroy, 1997; He and Ohara, 2017). This process can influence the
snow accumulation rate and the formation/preservation of the isotopic record and other chemical signals (Bohleber, 2019;
Bohleber et al., 2013; Nakazawa et al., 2005). This adds complexity to dating and interpretation of ice cores archives,
particularly for those retrieved at high elevation in non-polar glacierized areas subjected to significant snow melt and
wind redistribution. In this case, annual layer counting is difficult because surface melt and/or wind redistribution remove
snow layers formed across months or seasons (Neff et al., 2012). As these processes are typical of these mid-to-low
latitude regions and are part of the glacial archive's response to climate change, their understanding is a fundamental
prerequisite for paleoclimatic reconstructions.
Glacier mass balance modelling at ice core drilling sites is useful to reconstruct the formation and preservation of glacial
archives and their ongoing changes due to atmospheric warming. Mass balance models can provide information on the
amount of surface melt, meltwater percolation, and magnitude of snow accumulation by precipitation and wind drifting.
Overall, these model outputs can help to detect and characterize events linked to i) snow layer formation, ii) snow layer
removal, and iii) snow layer modifications (e.g. warming, wetting, refreezing). This information may also help in
detecting deviations of ice core proxies from the generally assumed linear, univariate recording of local temperature
(Evans et al., 2013). In fact, this assumption may not hold in the long term, particularly under extreme conditions such as
current or past warm climatic phases, especially at sensitive locations such as mid-to-low-latitude glacierized areas.



Proxy system models (models that describe the processes by which environmental conditions are recorded in a proxy archive) linked to isotope-enabled atmospheric general circulation models (models that describe isotopic variations in precipitation for a geographic area) are increasingly used to constrain ice core-based paleoclimatic reconstructions and for complementing interpretations based solely on statistical analyses (Evans et al., 2013). Proxy system models of various complexities were developed for ice core proxy interpretation (e.g., Brönnimann et al., 2013; Hurley et al., 2016; Okazaki and Yoshimura, 2019). These models reconstruct how stable water isotopes are recorded in ice core archives, generating a pseudo proxy that is compared to the actual proxy, to complement it and improve its paleoclimatic interpretation. For example, the possibility to disentangle different processes (temperature, intermittency of precipitation, diffusion) affecting the isotopic records to extract the "real" climatic signal, by using a so called "virtual" ice core, has been studied for Antarctica (Laepple et al, 2018).

These models are optimized for sites where snow layer removal by melting is negligible and snow redistribution can be accounted for by stacking ice core records from adjacent sites located in the same area (e.g., Ekaykin and Lipenkov, 2009). Additionally, models implicitly assume stationarity (or negligible variations) of snow redistribution, melt, and meltwater percolation. This is not always the case, particularly for low-latitude drilling sites where cold climatic phases that are favourable for ice core proxy formation and preservation alternate with warm climatic phases that are unfavourable.

The Mt. Ortles drilling site, at 3859 m a.s.l. in the Eastern Alps (Italy), has been characterised by a rapid warming of the firn and snow layers since the 1980s. Atmospheric warming is seriously threatening this paleoclimatic archive (Gabrielli et al., 2010) due to the increasing length and intensity of the ablation season, causing significant surface melt and meltwater percolation. In summer the firn layer is now entirely isothermal at the pressure melting point (Carturan et al., 2023). The ice below the firn-ice transition (30 m depth) is still cold, with temperature down to -2.8°C close to the bedrock in 2011 (Gabrielli et al., 2012). This shows that conditions favourable for glacial archive formation and preservation still exist on Mt. Ortles. However, the relatively low elevation of this ice core-drilling site makes it sensitive to climatic fluctuations and particularly vulnerable to non-linear processes controlled by snowmelt, snow metamorphism, and wind redistribution.

In this paper, we present a model-based reconstruction of the mass balance history and firn stratigraphy at the Mt. Ortles drilling site in the 1996-2011 warm period, including the extremely warm summer 2003 (García-Herrera et al., 2010). The aim is understanding the effects of exceptionally warm periods on glacier mass balance and the ice core paleoclimatic archive. Specifically, the mass balance model used in this work was implemented to: i) model the formation of snow and firn layers, ii) identify snow layers removed by ablation, iii) reconstruct the air temperature during the formation of snow layers that survived successive ablation. This latter is a pseudo proxy that we ultimately compare to stable water isotopes retrieved in the firn layers of the same period, to revise the dating of the firn portion of the Mt. Ortles ice core.

## 2 Study area

Mount Ortles (3905 m a.s.l.), in the Ortles-Cevedale Mountain Group, is the highest summit of South Tyrol in the Eastern European Alps. Its northern flank is covered by the Alto dell'Ortles Glacier (Oberer Ortlerferner-Vedretta Alta dell'Ortles), which extends over an area of 1.06 km$^2$ (2017) and ranges in elevation between 3018 and 3905 m a.s.l. (Fig. 1). The ice core drilling site is located at 3859 m a.s.l. in the upper accumulation area, close to a saddle (Figs. 1 and 2).



The glacier's maximum thickness at the drilling site is about 75 meters (Gabrielli et al., 2012) and the ice is ~7-kyr old at
the glacier base (Gabrielli et al., 2016). In this zone the glacier is polythermal, with temperate firn and cold ice underneath
the firn-ice transition, at ~30 m depth (Gabrielli et al., 2012). The Ortles ice archive is one of the only two cold ice archives
found in the eastern Alps, with the other being the nearby Weißseespitze (Cima del Lago Bianco) summit ice dome
(Bohleber et al., 2020; Gabrielli et al., 2012).
The local climate is characterized by a continental regime, with a mean annual precipitation in the period 1981-2010 of
800–950 mm yr$^{-1}$ at the valley floor in Solda (Adler, 2015). The annual precipitation on the top of Mt. Ortles is estimated
to range between 1300 and 1400 mm, using in-situ mass balance observations performed between 2009 and 2016,
(Carturan et al., 2023). This precipitation estimate is subject to large spatial variability due to the influence of wind on
snow accumulation and redistribution.
The mean annual air temperature at 3850 m a.s.l. on Mt. Ortles is about -9°C. On the glaciers of the Ortles-Cevedale
Group the snow cover follows a typical annual cycle, with accumulation prevailing between October and May, and
ablation between June and September. Due to the high elevation of the drilling site, snowfalls are also frequent during
summer. There is high interannual variability in the amount and duration of ablation events, which occur primarily during
heatwaves. Liquid precipitation is very rare, although some rain events have been recorded at the drilling site elevation
over the past 15 years (Carturan et al., 2023).








Figure 1: Location of the drilling site and automatic weather station (AWS) on Mt. Ortles. The background hill-shaded DEM (2017 lidar survey) is from http://geocatalogo.retecivica.bz.it/ (last access: 10 January 2025) (Agenzia per la Protezione civile, Autonomous Province of Bolzano).





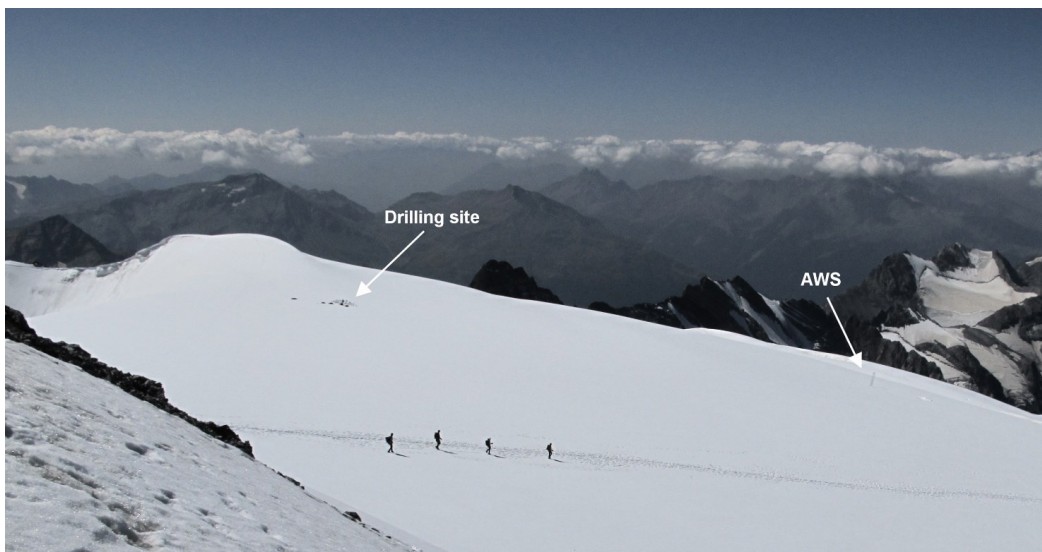


Figure 2: Photo of the upper accumulation area of Alto dell'Ortles Glacier taken from the summit of Mt. Ortles on 31
August 2015.


## 3 Methods

### 3.1 Ice core drilling operations

Four ice cores were drilled within 20 m of each other during September and October 2011 on a small col on the Alto
dell'Ortles Glacier, between the summit of Mt. Ortles and the Vorgipfel (UTM zone 32T, 618364 m easting, 5151531 m
northing, 3859 m altitude). In this study, we focus on core 1, which reached 73.53 m depth, because it is the only one for
which both complete isotope and pollen records are available. (Gabrielli et al., 2016).

142

### 3.2 Ice core analyses

Ortles core 1 was cut in a cold room (-20°C) at the Ca' Foscari University of Venice (Italy). Analysis resolution increased
with increasing depth, from 9 cm per sample (0-5 m depth), to 2 cm per sample (from 49 m to bottom).

Oxygen and hydrogen isotopic composition analyses were performed with two analytical methods. One method uses the
well-established $CO_2$-$H_2$/water equilibration technique (Horita et al., 1989) which couples an automatic equilibration
device (Finnigan MAT HDO 1086) with an isotope ratio mass spectrometer (Thermo-Fisher Delta Plus Advantage). Five
ml of water were used and the analytical uncertainty for $\delta^{18}O$ and $\delta D$ was ± 0.05‰ (1σ) and ±0.7‰ (1σ), respectively.
The other method was the wavelength-scanned cavity ring-down spectroscopy technique (PICARRO model L1102-i).
Since the injections of water samples can be affected by between-sample memory effects (Penna et al., 2012), samples



were injected (2 □l) 8 times and results were filtered using an outlier test. The analytical uncertainty for $\delta^{18}$O and $\delta$D was
± 0.10‰ (1σ) and ±0.5‰ (1σ), respectively. In each analysis run, two internal standards (periodically calibrated against
the IAEA international standards V-SMOW2 and SLAP2) were analysed along with the samples, and used for building
a calibration curve. The results were reported in the usual delta notation (δ) and expressed as per mil (‰).
Pollen analyses were performed at the Institute of Botany of the University of Innsbruck. Aliquots of up to 35 ml water
(depending on sample resolution) were used. Each sample was decontaminated with cold distilled water and the volume
of the water resulting from the ice melting was measured. Samples were processed with acetolysis (Erdtman, 1960) and
the pollen content was concentrated by hydro-extraction (centrifugation) and then prepared in glycerine slides (Faegri
and Iversen, 1989). The complete pollen content of the samples has been identified and quantified, as detailed in Festi et
al. (2015). For each sample, pollen concentration (grains ml$^{-1}$) was calculated. To detect the seasonality a principal
component analysis (PCA) has been performed on the pollen dataset according to the methodology developed in Festi et
al (2015). Three principal components indicative of the three main flowering seasons (spring, early summer and late
summer) were hereby extracted and are presented graphically. Peaks in component score values of a specific PC reflect
a pollen content characteristic predominant in the season corresponding to that particular PC. This method was applied to
the dataset as previous studies showed that the Ortles glacier pollen assemblages are representative for the regional
vegetation and comparable with airborne assemblages recorded at the nearby aerobiological stations (Festi et al., 2015).

**3.3 Mass balance observations**
Seasonal and annual glacier mass balances were measured at the drilling site and at the automatic weather station site
(Section 3.4., Figs. 1 and 2) from June 2009 to September 2014. Winter balance observations were typically carried out
in June/early July before the onset of melt, while summer/annual balance were performed in late August/early September
at the end of the melt season.
Observations consisted of snow depth soundings with a metal probe in the surroundings of the two sites, and of snow/firn
density measurements inside snow pits dug to the previous summer surface. Detailed snow stratigraphic observations
were carried out at shaded snow pit walls, comprising snow/firn temperature, hardness, grain type and size, location of
ice lenses and dust layers (e.g. Gabrielli et al. 2010). Stratigraphic observations were helpful in recognising summer
surfaces both in snow pits and during snow depth soundings. Density measurements were used to convert snow depths
into water equivalent depths.

**3.4 Meteorological observations**
The meteorological data used in this work are from an automatic weather station (AWS) located in the valley floor village
of Solda (Fig. 1, 1907 m a.s.l.) about 4.5 km northeast of Mt. Ortles. This AWS is part of the network of AWSs operated
by the Hydrological Office of the Autonomous Province of Bolzano (meteo.provincia.bz.it).
To collect additional meteorological observations, the Ortles paleoclimatological project (ortles.org) installed an AWS
close to the drilling site in October 2011 (Figs. 1 and 2). The AWS worked until June 2015 and was equipped with air
temperature, relative humidity, wind speed and direction, shortwave and longwave incoming and outgoing radiation, and
snow depth sensors. Details on Ortles AWS instrumentation and datasets are provided in Carturan et al. (2023).




## 3.5 Mass balance modelling

The mass balance model used in this study is EISModel, an energy-index model implemented for mass balance
computations on glaciers and seasonal snowpacks. The model in its original version is described by Cazorzi and Dalla
Fontana (1996), followed by Carturan et al. (2012a) who presented an advanced version for glacial environments. The
model was further developed for applications on Mt. Ortles, as detailed in Festi et al. (2017). In this section, we recall the
main features of EISModel and describe how it was applied to the study area.
The model was applied at hourly time steps. Snow accumulation was calculated from the hourly precipitation data of the
Solda AWS, validated against other neighbouring AWSs (Madriccio at 2825m and Cima Beltovo at 3328 m) and corrected
for gauge under-catch errors using the method proposed by Carturan et al. (2012b). Precipitation was extrapolated to the
elevation of the study site using a precipitation linear increase factor (PLIF, % km$^{-1}$), which is a lumped parameter that
accounts for the vertical increase of precipitation with elevation, preferential deposition, and erosion by wind.
Precipitation was classified as liquid or solid depending on the hourly air temperature, which is extrapolated from the
Solda AWS using monthly-variable temperature lapse-rates calculated between the Solda and Ortles AWSs.
Hourly melt rates were calculated using the following equation:
$$MLT_t = RTMF \cdot CSR_t(1 - \alpha_t) \cdot T_t \tag{1}$$
where $RTMF$ is the radiation–temperature melt factor (mm h$^{-1}$ °C$^{-1}$ W$^{-1}$ m$^2$), $CSR_t$ (W m$^{-2}$) is the clear-sky shortwave
radiation computed hourly based on the local topography, $T_t$ is the air temperature, and $\alpha_t$ is the surface albedo calculated
in function of cumulative positive $T_t$ (Carturan et al., 2012a).
The RTMF and PLIF parameters were initially calibrated at the Ortles AWS site using mass balance observations carried
out between 2009 and 2014. For a more robust calibration, we extended backwards to 2005 the mass balance observations,
using pollen dating of firn layers (Festi et al., 2017). We preferred to initially calibrate the model at the AWS site, rather
than at the drilling site, because field observations are more extensive and consistent at that site. The PLIF parameter was
then recalibrated at the drilling site, to account for lower snow accumulation. Finally, the model was applied to reconstruct
the cumulated mass balance in the period from September 1996 to September 2011.
The pseudo proxy to be compared to stable water isotopes in firn cores was obtained by calculating the air temperature
during the formation of snow layers that survived to ablation. We define as a snow layer the water equivalent of snow
that accumulates at the surface in one hour. For each snow layer surviving the following ablation, the model provided its
time and date of formation and the air temperature during its deposition, which is a variable named SLFT (snow layer
formation temperature). The model did not explicitly simulate snow removal by wind drift, which was computed
statistically by means of the PLIF multiplicative factor. The water equivalent thickness of each snow layer composing the
final snowpack was finally adjusted to account for layer thinning with depth, using the Nye (1963) ice flow model
(similarly to, for example, Eichler et al. (2000) and Brönnimann et al. (2013)).

## 4 Results

## 4.1 Model calibration



The calibrated values of the two parameters RTMF and PLIF optimized at the AWS site on Mt. Ortles were $10^{-3}$ mm h$^{-1}$
°C$^{-1}$ W$^{-1}$ m$^2$ and 15 % km$^{-1}$, respectively. At the ice core drilling site (located 200 m uphill of the AWS), the PLIF was
recalibrated to 8 % km$^{-1}$ to account for lower snow accumulation, probably due to higher wind erosion.
The model performance, expressed by the root mean square error (RMSE), is better for summer balance compared to
winter balance, and in closer agreement with observations at the AWS site, compared to the drilling site (Fig. 3). This
behaviour was expected because the drilling site is more exposed to wind action and likely experiences stronger snow
redistribution, particularly during winter. These results are similar to previous applications of EISModel on Mt. Ortles
(Festi et al., 2017).

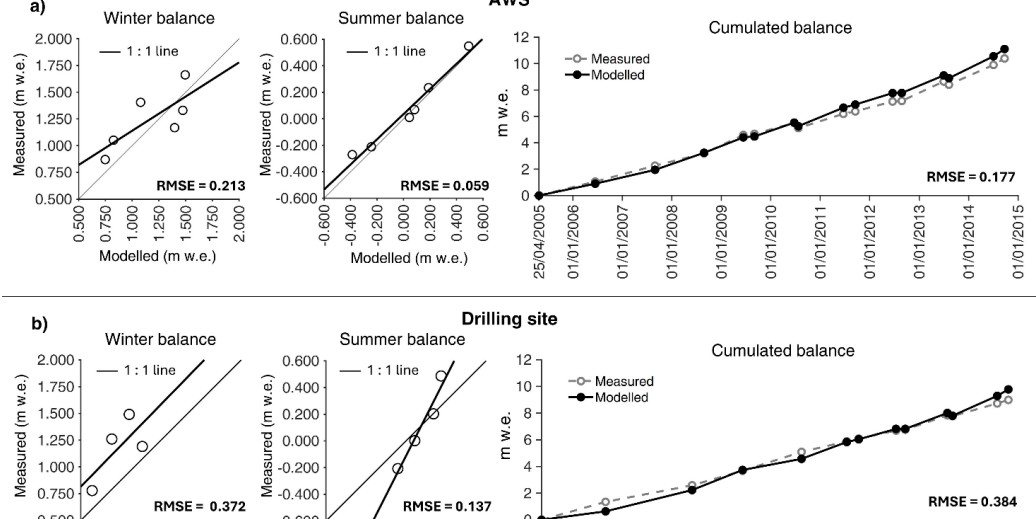

Figure 3: EISModel calibration results obtained on Mt. Ortles at a) the AWS site and b) the drilling site, for winter,
summer and cumulated mass balance, in the period (2005-2014).

**4.2 Mass balance behaviour**
Based on EISModel calculations, 21.4 and 18.9 m w.e. accumulated from September 1996 to September 2014 at the AWS
and ice core drilling sites, respectively (Fig. 4). Accordingly, the accumulation rate averaged 1.18 m w.e. y$^{-1}$ at the AWS
site and 1.05 m w.e. y$^{-1}$ at the drilling site. The accumulation rate was smaller between 1997 and 1998 and between 2003
and 2008, and increased in the periods between 1999 and 2002 and after 2008 (Figs. 4 and 5).

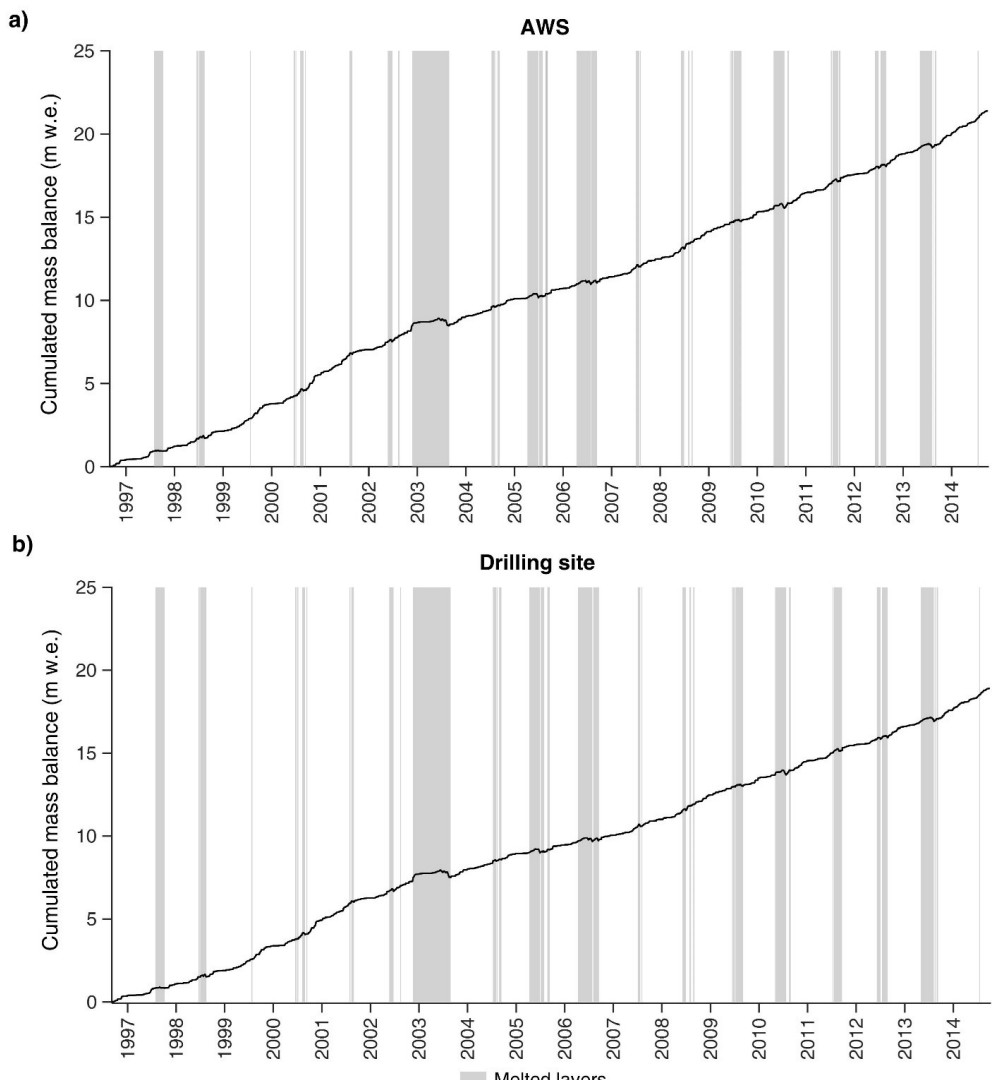

Figure 4: Cumulated mass balance modelled by EISModel at a) the AWS site and b) the drilling site. The vertical bars represent periods of snow accumulation that were removed by melt: large bars mean melting of snow accumulated over long periods (several months), whereas thin bars mean melt of snow accumulated over short periods.

The interannual variability of mass balance was remarkable at both sites (Fig. 5). The annual balance was closely correlated with the winter balance (r = 0.80) and slightly less correlated with the summer balance (r = 0.78). The summer balance was +0.25 m w.e. on average, but was close to zero in 2005, 2006 and 2013, and negative only in 2003 (-0.30 m w.e.).





The total melt (Fig. 5b) was also highly variable from year to year, and ranged between 0.19 m w.e. in 1999 and 0.76 m
w.e. in 2003. A few phases of intense and prolonged melt in 2003, 2005, 2006, 2009, 2013 and 2010 led to the removal
of snow layers accumulated over long periods (Fig. 4). According to EISModel calculations, the snow accumulated
between 17 November 2002 and 27 August 2003 (more than 9 months) was entirely melted during the 2003 European
heat wave. Five months of snow accumulation were removed in 2006 (from early April to early September), 4 months in
2005 (from early April to July), 3 months in 2013 (from May to early August), 3 months in 2009 (from June to August)
and 3 months in 2010 (from May to July).
The two years with best preservation of accumulated snow were 1999 and 2014, when melt was scarce (0.195 and 0.253
m w.e., respectively) and discontinuous. In these cases, melt removed only short periods of snow accumulation during
summer, without affecting the older layers underneath.





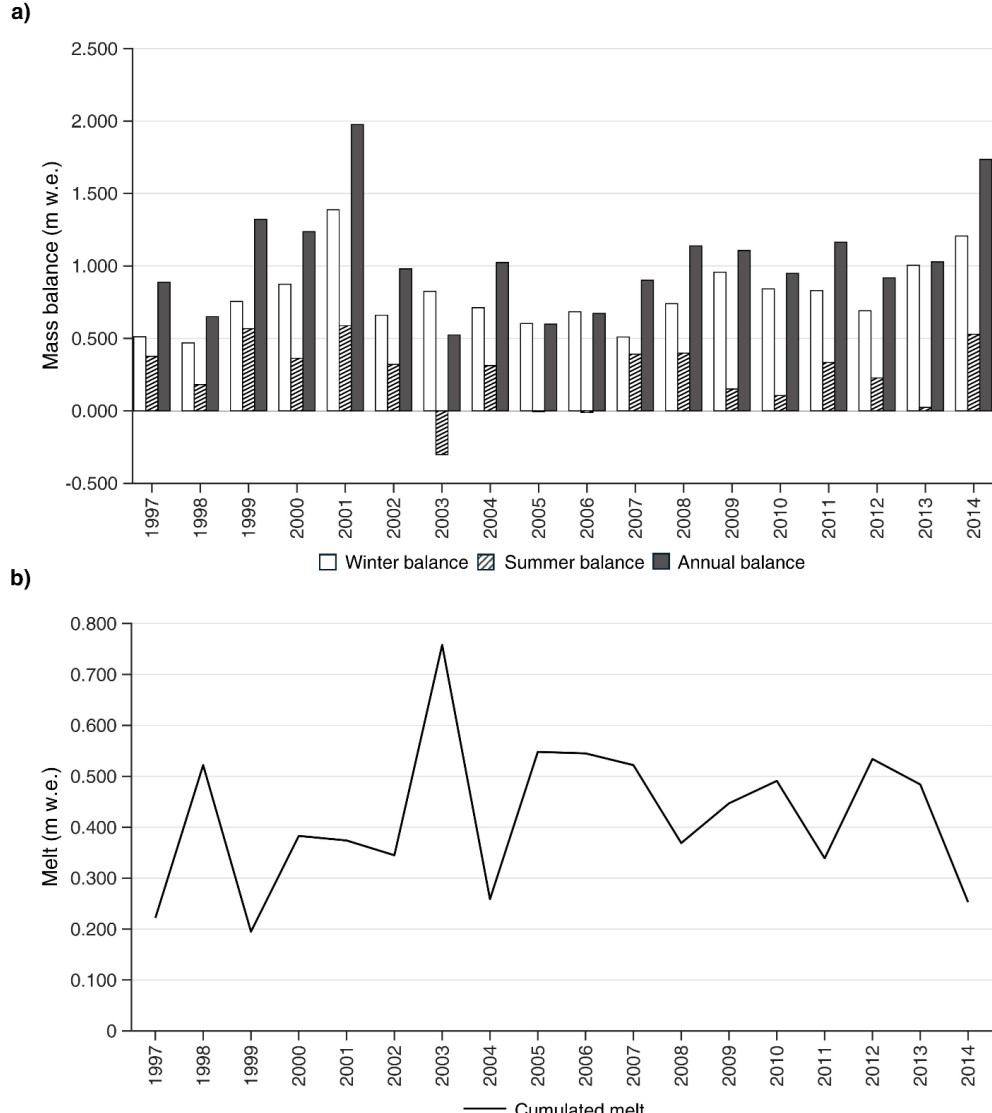

Figure 5: Interannual variability of a) seasonal and annual balance, and b) annual cumulated melt based on EISModel calculation at the Ortles drilling site.

**4.3 The modelled pseudo proxy**

In the analysed period, the pseudo-proxy SLFT series calculated by EISModel shows high interannual variability in the seasonality (i.e. the difference in SLFT between winter and summer) and in the accumulation of firn layers preserving a paleoclimatic signal (Fig. 6).



According to the model, during the years 2009-2011 snow accumulation largely offset ablation in both winter and
summer, preserving a marked seasonality in the water stable isotopes signal, with a good amplitude. Much less winter
snow accumulated in the two years 2007 and 2008, as can be seen from the thinner grey bars in Fig. 6. This caused
narrower troughs in SLFT, especially for 2007, whose trough is also higher compared to the following years. This is due
to scarce snow accumulation in the coldest months of winter 2007.
In 2006, summer ablation was 30% larger than average and removed the snow deposited from April to August.
Nevertheless, a large seasonal variation in SLFT is preserved thanks to snow accumulation in the coldest part of winter
and in late summer. A very sharp transition between cold and warm SLFT is observable in 2006 because winter and
summer layers are in direct contact.   In 2005, the cold-season trough in SLFT is barely visible because snow accumulation
in winter was scarce and spring snow was almost completely removed by melt.
The year 2004 shows a marked seasonality and a well-defined winter trough (comparable to that of 2006) thanks to good
winter accumulation and low summer ablation (Fig. 5). The two years 2002 and 2003 look like a single year, because the
exceptionally warm summer 2003 completely removed the snow accumulated in the winter season 2002-'03. The winter
2003 SLFT trough is therefore entirely missing from the record, and the November 2002 layers are in direct contact with
those of late August 2003 (thin white band between the 2003 and 2004 cold-season grey bars in Fig. 6). This is important
when counting annual layers and establishing a chronology in the firn core (see discussion below). Winter snow
accumulation was almost absent in the hydrological year 2001-'02, causing the formation of a rather warm cold-season
trough in SLFT.
High snow accumulation and low summer melt occurred in the 1999, 2000 and 2001 balance years. 2001 was a record-
setting winter accumulation year in this geographic area (Armando et al., 2002), as can be seen by the width of the cold-
season grey bar in Fig. 6. Summer accumulation was also the highest in the analysed period (Fig. 5). In 1999, there was
high summer accumulation, combined with negligible ablation, thus making it the second highest summer balance of the
analysed period.
In 1998 and especially in 1997, the seasonal variation of SLFT declines due to the very low accumulation in the coldest
months (lowest winter balance of the entire series, Fig. 5) and to the removal of the 1997 summer snow layers caused by
an anomalous late-summer melt event, between August and September.




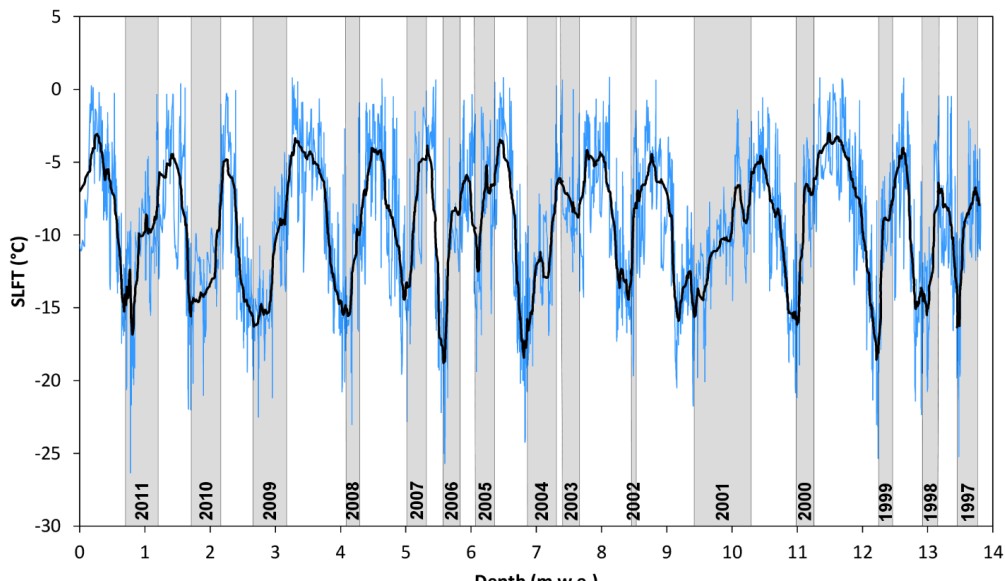


Figure 6: Snow layer formation temperature (SLFT) modelled at the drilling site by EISModel from September 1, 1996 to September 30, 2011. The black line is a 100-order centred moving average. The grey bars represent months from October to January (when pollen is not released, Fig. 7), in order to facilitate the following comparisons. The year is referred to the month of January (e.g., winter 1996-'97 is indicated as '1997').

302

### 4.4 Timescale based on pollen and stable isotopes

A tentative dating of the firn core extracted on Mt. Ortles was carried out based on annual layer counting based on stable isotopes and pollen measurements in the firn core (Fig. 7). We considered the depth interval from zero to 14 m w.e., the same as shown in Fig. 6. Winter layers were assigned based on troughs in the stable isotope series combined with very low/zero values in pollen concentration and PCA component scores (Section 3.2). Peaks in pollen concentration reflect the flowering season (Spring to Summer), while the lack of pollen indicates the non-flowering season (Autumn-Winter). Within each year, peaks in PC components indicate the presence of pollen types deriving from plant species blooming during spring, early summer and late summer (Festi et al., 2015 and 2017). This initial tentative timescale should represent 'routine' annual layer counting obtained using only experimental ice core data (like, for example, in Andersen et al., 2006; Takeuchi et al., 2019; Sinnl et al., 2022), independently from meteorological data or glacier mass balance observations or models.

The isotopic record is well preserved in the most recent four years (2011-2008) and is clearly smoothed by meltwater percolation before 2008, supporting the conclusions presented by Gabrielli et al. (2010). Stable isotope and pollen peaks match well within the firn layers. The pollen seasonality is well preserved for most years, except for 1999, 2006 and 2007, where the signals from spring, early summer, and late summer overlap. According to this initial tentative dating, the snow accumulation rate was larger before 2006 and smaller afterwards. When considering the layers with smoothed isotopes, there is a distinct peak that could initially be attributed to summer 2003; however, this is in clear contrast with EISModel



calculations, which indicate a complete ablation of the 2003 summer layers and removal of the associated isotopic signal
(Section 4.3).

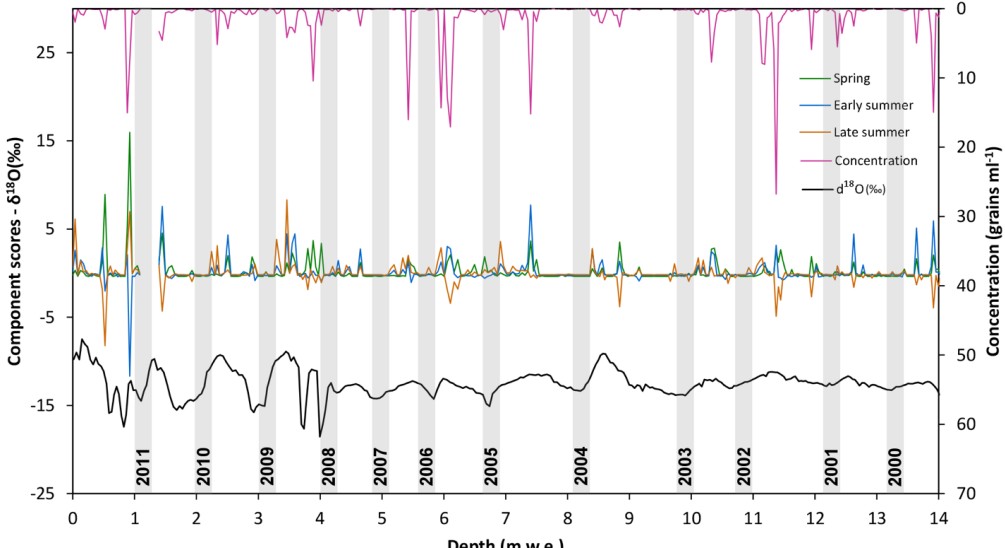


Figure 7: Concentration and principal component scores representative of spring, early summer and late summer of pollen
data extracted from the Mt. Ortles core 1, compared with the $\delta^{18}O$ values determined in the same core. Grey bars represent
mid-winter months (December and January) based on pollen data, and have arbitrary width. Annual layer counting is
based on $\delta^{18}O$ and pollen evidence.

**4.5 Refined dating using the modelled pseudo proxy**
A refinement of the tentative timescale in Fig. 7, based on the SLFT pseudo proxy modelled by EISModel, is presented
in Fig. 8. The annual layer counting from Fig. 7 matches the SLFT-based layer counting from 2011 to 2006. Below these
layers, we assigned the $\delta^{18}O$ trough at ~6.8 m w.e. depth to winter 2004 (instead of 2005) and considered the ~1 m w.e.
between 5.8 and 6.8 m as the result of two years of snow accumulation, 2004 and 2005. According to this interpretation,
the missing trough in the winter 2005 isotopic record is due to the low winter accumulation (Fig. 5a) and to the removal
of spring snow by ablation (Section 4.3, Fig. 6).
Similarly, we reinterpreted the $\delta^{18}O$ trough at ~8.2 m w.e. depth as winter 2002, whereas the winter trough of 2003 is
absent in the isotopic record because those winter layers completely melted during the 2003 warm summer. The annual
counting of firn layers below 2002 was consequently shifted according to these considerations, without other changes
compared to Fig. 7.
This reinterpretation (Fig. 8a) presents limited chronological discrepancies compared to the SLFT pseudo proxy (Fig. 8b),
without systematic under/overestimation of accumulation rates. These discrepancies would cancel each other over the
depth/period considered, as can be noted from the black lines that connect Figs. 8a and 8b, whose tilt seems randomly



distributed. Interestingly the major peak in isotopes at ~8.5 m w.e. depth dates summer 2001, instead of 2003 as attributed
merely on isotopes and pollen data (Section 4.4). This peak matches with the highest summer accumulation in the analysed
period (Fig. 5a; Armando et al., 2001). The second highest summer accumulation year (1999) matches with the secondary
peak in isotopes at ~11.5 m w.e. depth.

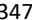


Figure 8: a) Same as Fig. 7 except annual layer counting reinterpreted based on b) snow layer formation temperature
(SLFT) modelled at the drilling site using EISModel. In a) the first timescale (shown in Fig. 7) is reported with a grey





font, to ease comparison. The two darker grey bars in a) indicate winter seasons that were missed in the timescale based
only on isotopes and pollen information, and added in the following using SLFT.

## 5 Discussion

The underlying assumption of our modelling approach is that the variability of local 2 m air temperature during snowfall
at the drilling site (i.e. the EISModel output variable SLFT) is representative of the $\delta^{18}$O variability in snow deposited at
the same site. This is equivalent to assuming a linear relationship between 2 m air temperature and $\delta^{18}$O, whose variability
is affected by other processes such as the origin and history of the water vapour in the air mass, condensation cycles, sub-
cloud humidity, and preferential deposition and redistribution of snow by winds (Dansgaard, 1964; Sturm et al., 2010).
Another assumption is that the original $\delta^{18}$O variability after snow deposition was preserved on Mt. Ortles, which in realty
is most likely modified by post depositional processes such as melt, sublimation, condensation, diffusion, isotopic
exchange between atmospheric water vapour and snow, and percolation of melt/rainwater (Eichler et al., 2001, Steen-
Larsen et al., 2014). All these effects may have caused a post-depositional increase in $\delta^{18}$O values.
Because so many factors affect the original isotopic composition of snow and its variation through time, there are likely
limitations for our approach. However, the development of EISModel and of the pseudo proxy of snow temperature
formation were not intended to accurately reproduce the $\delta^{18}$O variability measured in the ice cores retrieved on Mt. Ortles.
Instead, it was designed to be a tool to improve the interpretation of paleoclimatic data preserved in snow and firn cores,
particularly through more accurate firn annual layer counting. The modelling approach is based on scientific literature
supporting the main assumptions that i) ice cores contain temperature information that can be extracted from stable water
isotopes (e.g. Brönnimann et al., 2013; Hurley et al., 2016, Steiger et al., 2017) and ii) this temperature information and
its seasonal signature are preserved (although smoothed) after limited meltwater percolation (e.g. Moser et al., 2024).
However, completely melted layers cannot be represented in the retrieved stable isotope record, and the EISModel
specifically accounts for that.
The EISModel is an intermediate-complexity model aiming at representing the dominant processes affecting glacier mass
balance and paleoclimate proxy formation/preservation, while requiring only a few meteorological input data
(precipitation and air temperature). According to Evans et al. (2013), best-compromise models for paleoclimatic
applications are more complex than univariate-linear in their response to environmental forcings, without the need to
capture all fine-scale processes at play in a given proxy system.
At the Mt. Ortles drilling site, the two major processes that likely influence the formation and preservation of the isotopic
record are i) the seasonal distribution of precipitation/accumulation (taking into account the seasonal wind erosion) and
ii) the intensity and duration of summer ablation. Both show high interannual variability (Fig. 5), which is typical of the
climate of this region. Due to their importance and strong variability in time, they are explicitly modelled by EISModel
to build a pseudo proxy (SFLT) that can be helpful in studying ice core records retrieved at similar sites.
Overall, considering the model used and the characteristics of the study area, the model's skill in reproducing observed
mass balance is satisfying (Fig. 3) because the magnitude of RMSE values is comparable to the typical errors in mass
balance measurements (Zemp et al., 2013). A major simplification of our model is the use of a linear relationship between
accumulation and precipitation, meaning time-invariant vertical precipitation gradient and wind redistribution. This is



common in glacier mass balance models of similar complexity and is unavoidable without additional in-situ direct
observations of precipitation and snow redistribution.
The impact of this simplification is visible in the simulations of winter balance, which display a higher RMSE compared
to summer balance (Fig. 3). However, it must be considered that uncertainty and high spatial variability also affect winter
balance field measurements, not only simulations. Differences up to 0.5 m in snow depth above previous year's summer
surface were measured within soundings that were just 10 m apart. This is exacerbated by the difficulty of identifying the
summer surface in snow pits and snow depth soundings. However, since the RMSE does not exceed one third of the
annual snow accumulation rate, we are confident in the overall model's skill to discriminate between high and low
accumulation years/seasons.
Our approach share some similarities to that used by Brönnimann et al. (2013) who replicated the ice core from the
Grenzgletscher (Switzerland, 4200ma.s.l.) on a sample-by-sample basis by calculating precipitation-weighted
temperature (PWT) over short core intervals. However, this approach did not account for melt, whose effects are instead
explicitly calculated by our glacier mass balance model. Considering the increasing air temperature, melt events are
increasingly affecting high-altitude regions of temperate mountain areas, resulting in strong alterations of glacier mass
balance and vertical shift of dry-snow/percolation/wet-snow zones on glaciers. Besides deterioration of paleoclimatic
information contained in ice cores due to meltwater percolation, exceptional melt events can physically remove months
of snow accumulation from glaciers. This is happening at increasingly high elevation in the European Alps (Baroni et al.,
2023; Carrer et al., 2023) and is no longer limited to drilling sites below 4000 m a.s.l. (Huber et al., 2024). Even though
current melt rates are extremely high, similar intensity of melt might have occurred also in past epochs, for example
during the Holocene thermal maximum (Renssen et al., 2012; Kalis et al., 2003).
For these reasons, embedding a glacier mass balance model in proxy system models is a useful approach because of the
importance of ablation, together with accumulation, in determining how the climatic signal is recorded in the ice core
paleoclimatic records. Even though ablation may be assumed as negligible at an ice core drilling site at the present time
under the current climate, it might have been significant in the past, particularly at glacial-interglacial time scales.
Therefore, testing how melt events impact ice core records by means of glacier mass balance models embedded in proxy
system models can be useful for improved dating and interpretation of these paleoclimatic archives.
The SLFT pseudo-proxy clearly adds robustness to the firn core chronology, because it explicitly highlights the
interannual variability of snow accumulation and ablation. Nevertheless, the interpretation of some features in the stable
isotopes and pollen records remain uncertain. For example, the $\delta^{18}O$ peak at ~ 8.5 m w.e. (Figs. 7 and 8) is anomalous,
considering that the records from this section of the core were smoothed by melt water percolation. In Section 4.5 we
tentatively attributed this peak to the very high summer accumulation in 2001 (Fig. 5), but there might be alternative
explanations.
For instance, post-depositional effects due to the 2003 European heat wave could have caused an increase in $\delta^{18}O$ values
of the snow layers re-exposed during this extreme heat event, similarly to what has been observed for post depositional
changes of the isotopic composition of surface snow on the Greenland ice sheet (Steen-Larsen et al., 2014). In 2003, on
Mt. Ortles, the strong percolation of melt water might have relocated pollen grains (vertical and/or lateral drainage) thus
explaining the relative scarcity of pollen in the 2001 and 2002 layers in Fig. 8a.



The lack of a distinct peak in this section of the SLFT series, similar to the peak at ~ 8.5 m w.e. observable in the δ18O
series, might suggest that the latter depends on post-depositional effects that are not considered in the ESIModel, like
sublimation, condensation, diffusion, and isotopic exchange between atmospheric water vapour and snow (Sokratov and
Golubev, 2009; Steen-Larsen et al., 2014; Ebner et al., 2017; Madsen et al., 2019).
As already discussed, the SLFT pseudo-proxy does not account for snow redistribution by wind, which is an important
process at this high-elevation site exposed to strong winds. According to snow depth observations on Mt. Ortles and
similar locations in this region (e.g., Fischer et al., 2022; Carturan et al., 2023) wind erosion strongly prevents snow
accumulation in the colder winter months, between January and March. For this reason, we think that further
improvements in the development of the pseudo proxy might be possible by including a simple parameterization of snow
erosion and its dependence on air temperature (Li and Pomeroy, 1997; He and Ohara, 2017).

**6 Conclusions**
In this paper, we present a model that simulates the mass balance history and reconstruct the glacier stratigraphy at the
Mt. Ortles ice core drilling site between 1996 and 2011. The model calculates the air temperature during the formation of
snow layers (SLFT). The SLFT is used as a pseudo proxy for improved dating and interpretation of the ice core
paleoclimatic archive retrieved on Mt. Ortles in 2011.
The model demonstrates good skill in reproducing the observed mass balance and proves to be useful for the interpretation
of the ice core data. It is particularly valuable in detecting two major ambiguities in annual layer counting based on stable
water isotopes and pollens, namely the two years 2005 and 2003, which lack a winter signal in the isotopic record. Without
the model reconstruction of the local mass balance, it would not have been possible to identify and quantify these two
anomalies, which stemmed from melt-induced removal of snow layers accumulated over several months or seasons.
Considering the current rate of atmospheric warming and the impact of extreme melt events (such as the warm 2003
summer in the European Alps), we suggest that modelling approaches accounting for accumulation and ablation processes
can be useful for understanding how the paleoclimatic signal is formed and preserved in ice cores.
These considerations may be valid for both the current warming phase and past climatic changes. Dating and interpretation
of ice core records formed during the Holocene thermal maximum, for example, may present issues similar to those
highlighted in this paper. During that period and perhaps in other warm phases of the Holocene (Renssen et al., 2012),
above-average summer melt, melting of large quantities of snow at the surface, and variations in snow drifting likely
occurred. Model-based studies similar to the one presented in this study can provide insights into these processes and can
enable detection of these events in past climate reconstructions based on ice cores, in particular those obtained near the
lower altitude limit for preserving atmospheric signals in snow and ice layers.

**Data availability**
Data are available from the corresponding author upon reasonable request.



**Author contributions**

LC designed the methodological approach. PG, LC, RS, GDF carried out the fieldwork. FDB and TLZ processed the meteorological data. DF and KO performed the pollen analyses. PG, GD and BS performed the isotopic analyses. FC wrote the EISModel and implemented the SLFT pseudo proxy. AI and TLZ calibrated and run the EISMODEL. LC prepared the first draft of the manuscript with contributions from PG, TLZ, AI, BS, and GD. All authors contributed to the editing of the manuscript.

**Competing interests**

The contact author has declared that none of the authors has any competing interests.

**Acknowledgments**

The authors are grateful to all the students, technicians and scientists who contributed to the field activities in the period from 2009 to 2016; the alpine guides of the Alpinschule of Solda; the helicopter companies Airway, Air Service Center, Star Work Sky; and the Hotel Franzenshöhe for logistical support. The authors acknowledge the editor and reviewers for their comments and suggestions.

**Financial support**

The research was funded by the Italian MIUR Project (PRIN 2010-11), "Response of morphoclimatic system dynamics to global changes and related geomorphological hazards" (local and national coordinators are Giancarlo Dalla Fontana and Carlo Baroni) and was carried out within the RETURN Extended Partnership and received funding from the European Union Next-GenerationEU (National Recovery and Resilience Plan – NRRP, Mission 4, Component 2, Investment 1.3 – D.D. 1243 2/8/2022, PE0000005). The core samples were obtained as part of the Mt. Ortles Ice Core Project funded by: NSF Awards 1060115 and 1461422 with the logistic support of Ripartizione Protezione antincendi e civile of the Autonomous Province of Bolzano in collaboration with the Ripartizione Opere idrauliche e Ripartizione Foreste of the Autonomous Province of Bolzano and the Stelvio National Park. This is Ortles project publication 13 (www.ortles.org).

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
