# Peer review of "Reconstruction of mass balance and firn stratigraphy during the"

_EGUsphere, 2025_

## Author Response (AR2)

Dear Editor,

we would like to thank you and the two Reviewers for the reviews and suggestions that helped us improving our manuscript.

Based on your suggestion, we uploaded the datasets used in this work on Zenodo, and provided the digital object identifier in the section "Data availability" at the end of the manuscript.

We have addressed all the points highlighted by the reviewers and we modified the manuscript accordingly. In particular, we added a discussion in Section 5 regarding the percolation of meltwater, as suggested by Reviewer 2.

In the following, we answer in detail to the specific comments made by the reviewers. The author responses are reported in blue colour right below the reviewers' comments. Line and page numbers indicated by the reviewers are referred to the submitted paper, whereas our replies refer to the revised version of the paper (track-change version).

**Reviewer 1**

The authors present a novel approach to surmount the challenge of interpreting altered paleoclimate records such as the ice cores recovered from Mt. Ortles. This is a worthy pursuit, as recovering paleoclimate information from fast changing polar and alpine regions must be done before information is obscured outright by increasing temperatures and increased ablation and meltwater alteration of chemical snow and ice stratigraphy.

Carturan et al make a useful advance in understanding the Ortles records, where some time periods in the core are missing due to less snow accumulation and/or increased melt. The approach models the mass balance history and firn stratigraphy at the site from 1996-2011 to reproduce stratigraphy observed in the firn/ice core records and check interpretation of those samples against their model based expectation of what is preserved in the stratigraphy. The model approach appears well calibrated against accumulation as observed at an automated weather station and the drilling site, despite considerable inter annual variability in mass balance.

This modeling approach importantly provides a somewhat independent verification of the annual layer counting based on water stable isotopes and pollen concentrations. This will help going forward in considering uncertainty in the climate interpretation of the pollen and water isotope data, giving some sense of uncertainty due to dating error.

I have few direct concerns about the approach and manuscript, and indeed consider it a valuable contribution to the discipline that we'd do well to apply to similar sites such as our work at Mount Waddington, British Columbia, Canada.

-Peter Neff

Reply: we are grateful to the reviewer who finds the manuscript a valuable contribution and the procedure we propose useful and applicable to his study sites in Canada.

**Reviewer 2**

In this study, the authors used their EIS model to record temperatures during snow accumulation within the model and compared this data with ice core information to provide additional information for reconstructing paleoclimate conditions. Considering the amount of melting, creating a simulated core with

temperature records during snow layer formation is scientifically useful research that complements actual ice core information. There are a few areas that need additional explanation or comment. Please consider these when you submit your revised paper.

Minor comments

L152 Possible character-encoding issue. Please verify that all special characters and symbols are displayed correctly.

Reply: thanks for spotting this, the manuscript has been corrected and checked throughout.

L205 The snowmelt calculation equation here is expressed as a product of radiation and temperature, but in general heat balance calculations, solar radiation and air temperature are added together to estimate snowmelt. According to the author's previous manuscript, Caturan et al (2012), EISmodel has an option to select additive equations. Is there a reason for using multiplicative equations in this study?

Reply:  we used the 'native' algorithm of EISModel (Cazorzi and Dalla Fontana, 1996), which demonstrated better performance in simulating summer balance and melt in previous applications (Carturan et al., 2012a). Melt simulation accuracy is relevant for this model application on Mt. Ortles.

Cazorzi F and Dalla Fontana G (1996) Snowmelt modelling by combining air temperature and a distributed radiation index. J. Hydrol., 181(1–4), 169–187.

L214-221   SLFT states that redistribution is not included, but according to Carturan et al (2012) paper, the redistribution calculations are included in the EIS model. Was this redistribution calculation deliberately omitted from this analysis?

Reply: the SRF (snow redistribution factor) implemented in the EISModel accounts for the spatial variability of snow redistribution, not for its variability in time. Because on Mt. Ortles we apply the model to simulate the temporal evolution of the mass balance at single locations (AWS and drilling site), we are not interested in the spatial variability of snow redistribution. The SRF would be a constant for each of the two locations (SRF is a grid where each pixel has a constant SRF value, depending on local topography). For this reason, it makes sense to lump this factor in the PLIF multiplicative factor applied to precipitation data, which includes the two major processes regulating snow accumulation, i.e. the vertical precipitation gradient and the snow redistribution. We are aware that this is a limitation in our approach, as we write at L442-447 in the revised manuscript (L429-434 in the submitted manuscript). Clarified in the text (L218-219 of the revised manuscript).

L252-258 If the annual snowmelt is around 50 cm water equivalent, the effect of water infiltration is also present. Although the main focus of this paper is not to consider infiltration, it is expected that adding a discussion of the effects of infiltration somewhere will provide useful information. If the effects of infiltration can be added to the SLFT model, is there a possibility to remove these effects from the model or to estimate the reliability of the agreement between isotope concentrations and SLFT?

Reply: as suggested, we added a discussion in Section 5 (lines 386-396 in the revised manuscript) regarding the percolation of meltwater and the feasibility of including its effects on the pseudo proxy developed in our work. Given the complexity of meltwater infiltration processes and their effect on the isotopic records (still topic of research and discussion in the scientific literature), and considering the type of model we are using, we preferred to keep the pseudo proxy as simple as possible in order to reduce the possible sources of uncertainty.

L260-261 Even if the annual snowmelt is around 20 cm water equivalent, there is a possibility that water infiltration could affect old snow if there is a lot of snowmelt in one event during that season. Was there any evidence that old snow is not affected?

Reply: here we mean that the older layers underneath were not melted, we do not mean that they were not affected by percolation. Rephrased for clarity (L262-263).

L271-272 If this sentence refers to the isotope amplitude shown in Fig. 7, it would be better to include (Fig. 7).

Reply: here we are describing the model output variable SLFT (i.e. the pseudo proxy), and for this reason we only refer to figure 6 and not to figure 7 (which shows the real proxies). Rephrased for clarity (L274).

L304-305 Can you show the scale of seasonal pollen amount in the centre of Fig. 7?

Reply: the seasonal pollen amount in the centre of Fig. 7 is reported as principal component scores (left Y-axis). We have modified the figure caption to clarify.

L330-339 Although this is not mandatory, Fig. 8 is a combination of Figs. 6 and 7, so it would be better to combine them into a single figure.

Reply: in our opinion, even if there is some redundancy, it is better to keep these figure separated to ease understanding.

L417-419 I am not familiar with the behaviour of isotopes, but I have a question that comes to mind. If water is ponded here by a capillary barrier or something similar, is it possible that the isotope concentration will increase?

Reply: We would not expect relative isotope concentration changes (fractionation) to occur in the case illustrated by the reviewer as isotopic fractionation typically occurs when solid-liquid-gas phase changes are involved.

L420 δ appears to be garbled.

Reply: thanks for spotting this, the manuscript has been corrected and checked throughout.

L423-424 I think that pollen is a medium that is less susceptible to the effects of water percolation than soluble substances. Can you explain the author's opinion on the extent to which water movement affects pollen movement?

Reply: Even if literature generally reports that pollen has a low susceptibility to water percolation, there is the possibility that extreme melt events (like the 2003 one in the European Alps) might induce significant relocation of pollen. Ewing et al. (2014), for example, report substantial vertical and horizontal pollen transport during a controlled experiment, following snowmelt. Reference added in the text (L437 in the revised manuscript).